# Benefits, Facilitators and Barrier Reductions in Physical Activity Programmes for People with Severe Mental Disorder: A Systematic Review

**DOI:** 10.3390/healthcare11091215

**Published:** 2023-04-25

**Authors:** Cristina Méndez-Aguado, Adolfo J. Cangas, José M. Aguilar-Parra, María J. Lirola

**Affiliations:** 1Hum-760 Research Team, Department of Psychology, Health Research Centre, University of Almeria, 04120 Almeria, Spain; 2Hum-878 Research Team, Department of Psychology, Health Research Centre, University of Almeria, 04120 Almeria, Spain

**Keywords:** several mental disorder, physical activity, benefits, facilitators, barriers

## Abstract

The current high prevalence of people with Severe Mental Disorder and the high impact the latter has on their quality of life is one of the main problems in terms of health, as it affects both physical and mental health. One of the lines of action to intervene in these factors is the practice of physical activity, as this usually has a low level of participation due to different barriers. However, there are several facilitators that improve adherence to these practices. The main objectives of this work were (1) to conduct a systematic review of the scientific literature on the possible benefits obtained by people with SMD from their participation in physical activity programmes; (2) to identify the characteristics of physical activity programmes and determine the barriers to their implementation that have been considered and (3) the facilitators incorporated. To meet these objectives, the SCOPUS, Web of Science, PubMed, Dialnet and Elsevier online databases were consulted and, following the PRISMA statement, 17 articles were finally selected. Their analysis has revealed various physical, psychological and social benefits, as well as the barriers that appear in the intervention programmes, mostly related to personal factors and the programme itself, and those factors that facilitate their adherence or development, the most common being the carrying out of the activities outdoors, the inclusion of social components and the possibility of adapting the activities. In this way, the results obtained have made it possible to highlight the characteristics that should be taken into account when planning this type of intervention.

## 1. Introduction

Severe Mental Disorder (hereafter SMD) refers to psychopathological problems that have a long evolution (at least two years) and require support at various levels (such as social, educational, judicial, etc.). Formally, it includes the diagnoses of schizophrenia, bipolar disorder, major depressions and severe personality disorders [1,2]. Due to the high impact on the quality of life of people with these disorders and their current prevalence, they are considered to be a major health issue [3].

People with SMD are at increased risk of developing medical complications such as cardiovascular or respiratory diseases, metabolic syndrome, diabetes and obesity, among others, due in part to their lifestyle, eating habits and the adverse effects of medication or substance use [4,5]. They are more sedentary than the general population, spending more than 6 h a day sitting, thus not complying with WHO recommendations for aerobic physical activity at moderate or vigorous intensity between 150–300 min per week [6,7]. This lack of physical activity is related to the side effects of treatment and to the disorder’s own symptoms that contribute to demotivation [8,9].

They also stand out for having an unbalanced diet in which the consumption of discretionary foods, i.e., those high in energy but low in nutrients, predominating over the intake of fruit and vegetables [8]. This has a negative impact not only on their health but also on their psychological and social well-being [10]. Thus, their quality of life and life expectancy are diminished, the latter being between 10 and 20 years lower than the average for the rest of the population, with cardiovascular diseases being the most common cause of mortality [11].

Among the recommended practices to ameliorate the direct and indirect negative consequences of SMD is the practice of physical activity. Active participation in this type of activities, following WHO recommendations, has physical, psychological and social benefits [10,12].

Nevertheless, it is an activity with low adherence due to the barriers that this population encounters. On the one hand, there are personal factors, including emotional influences, which are mainly negative [13], lack of motivation [14], the symptoms of the disorder and the side effects of medication [15], as well as negative expectations about one’s own performance and the results to be obtained [16]. On the other hand, there are also certain social factors related to a lack of family and institutional support that have a negative impact on the level of motivation, as well as at the economic level [15]. Similarly, at the organisational level, there is a lack of staff responsible for the supervision and individual adaptation of physical activity programmes and the use of economic resources for their development and logistics [17]. Similarly, environmental constraints limit the spontaneity of activities and their adaptation to individual needs [18].

In this way, it is necessary to address the facilitators that contribute to reducing or lowering these barriers and, therefore, improving adherence to physical activity practice. To this end, considering the great importance of creating opportunities for practice and intervening on the motivation of participants, it is necessary for sporting proposals to have certain characteristics. These should be attractive, for example, by predominantly outdoor exercise [14], favouring the socialisation of those involved [19] and promoting inclusive programmes [9]. In addition, it is important that the activities are varied, combining both endurance and strength training, and their intensity should increase progressively and preferably not exceed moderate intensity [20,21]. Programmes should also have a certain degree of flexibility, being able to adapt to the abilities and needs of each individual in order to make them feasible and decrease the likelihood of frustration [22]. The development of autonomy [20] and the presence of support from the participants’ immediate environment, not only from the family but also from all those specialists working with them [23], are great facilitators, as well as proper advice on healthy habits [14].

However, despite recommendations to encourage physical activity, there are few programmes aimed at this population [24]. At the research level, the available studies stand out for considering the benefits in a fragmented way, not including the different physical, psychological and social factors on which to act. The same occurs when analysing the existing meta-analyses and systematic reviews, in which the benefits of these practices are presented, but either without differentiating between the factors mentioned above, focusing exclusively on them [6,25,26,27] or evaluating the level of motivation of the participants [14] and their level of physical activity [5]. Thus, there is a notable lack of studies in which the characteristics of interventions have been analysed in depth in order to identify good practices that counteract the barriers faced by this population, identifying the facilitators that are included in order to facilitate and promote adherence to these programmes.

With these considerations in mind, this research has three main objectives: (1) to review the scientific literature on the possible benefits obtained by people with SMD from their participation in physical activity programmes; (2) to identify the characteristics of physical activity programmes in order to determine which barriers to their implementation have been considered; and (3) to find the most important facilitators for adherence to physical activity programmes.

## 2. Method

### 2.1. Search Strategy

The systematic review carried out in this work was conducted in accordance with the PRISMA regulations [28,29]. The databases used were Web of Science (WoS), Scopus, PubMed, Dialnet and Elsevier, consulted during the month of June 2022. In these databases, “AND” and “OR” were used as truncation operators, establishing the search by keywords such as “physical activity” AND “severe mental disorder” and “physical activity” AND “severe mental disorder” OR “severe mental illness” OR “serious mental illness”.

### 2.2. Study Eligibility

The PICO analysis followed to determine the eligibility criteria:

P—Type of Participants: person with SMD, and type of studies: Quantitative studies (cross sectional or longitudinal);

I—Type of Intervention: intervention based on Physical Activity and measures evaluated;

C—Comparison: there was no need for a comparison group for the selection of articles;

O—Outcomes: reference to results, effects and barriers found in the physical activity interventions for persons with SMD;

Therefore, the following inclusion and exclusion criteria were applied.

Regarding the studies included in the review, we selected those written in both Spanish and English that (1) focused on people with a medical diagnosis of SMD, including the wide variety covered by this term, such as schizophrenia, bipolar disorder or personality and affective disorders; and (2) implemented a physical activity programme, evaluating the benefits obtained after its implementation. Exclusion criteria included studies that were not published as scientific articles, those to which only the abstract was available, those that did not meet criteria (1) and (2) mentioned above and those that did not belong to the 2012–2022 time range, established in order to access the most up-to-date information available, as this is a topic that has been less prominent in the past.

### 2.3. Publication Selection

To identify the relevant articles, the titles were first read. Subsequently, the abstracts were examined and, finally, the full texts were reviewed, and those that did not meet the pre-established inclusion criteria were rejected. Following the established search strategy, the initial number of publications obtained was 1298. However, once the titles had been analysed, the number was reduced to 49, of which 16 were finally selected after reading the abstracts and full texts. In the case of the selected articles, we extracted data related to the authors and year of publication, the type of study, the number of participants, the type of SMD considered, the variables measured, the characteristics of the intervention carried out, the results obtained, the barriers considered and the facilitators present.

## 3. Results

Once the research studies had been selected using the selection criteria, we proceeded to analyse the characteristics of each study following the PICOS method (see Table 1) and then summarised (see Table 2):

### 3.1. Type of Participants and Type of Studies

All participants were persons with Severe Mental Disorder, and all 17 articles were longitudinal in nature.

### 3.2. Intervention Modality and MEASURES Evaluated

The articles reviewed included interventions to observe and analyse the improvement of variables such as anthropometric measures, body composition, physical skills such as coordination, somatic parameters, quality of life, social interactions and medication dosage.

### 3.3. Comparison Interventions

Comparison interventions were carried out using different modalities of physical activity such as football or aerobic exercise, among others, in interventions ranging from 4 h to 10 months being the longest in time, with intervention studies typically lasting between 10–16 weeks.

### 3.4. Outcomes

The results varied according to the variables analysed, with significant improvements in all cases except for one study where no significant differences were found [32].

## 4. Discussion

The aim of this study was to identify the benefits obtained by people with SMD from their participation in physical activity programmes, as well as the main barriers and facilitators to this practice.

The results obtained have revealed benefits at different levels. On the physical level, the studies reviewed have shown improvements in anthropometric measures [30,31,35] and aerobic capacity [39]. These benefits improve functional exercise capacity [41] and reduce cardiometabolic risk [36]. This wide range of benefits derives from the type of approaches undertaken. When aerobic capacity has been the focus of improvement, the participants’ endurance and physical fitness have been the main focus [12,44] and when muscle strengthening activities have been carried out, strength and muscle mass, great allies of coordination and balance, have been worked on [45]. Likewise, the decrease in metabolic risk has also been highlighted by authors such as Piercy and Troiano [46], who have assessed this improvement with a sample of a population without SMD, showing that the benefits are obtained regardless of personal condition.

On the other hand, the practice of physical activity has also reported benefits at the psychological level. It has been found to intervene on quality of life directly [30] as well as on mastery of the environment [31] and reducing negative symptoms related to the mental disorder or the medication [33,38,40] while increasing self-esteem and autonomy [20]. These results are in line with the study conducted by Eime et al. [47]. This could also justify the results obtained in the study by Deenik et al. [34], who add a possible reduction in pharmacological consumption that would be related to the practice of physical activity and adherence to this type of programme, although the latter are results that have not been analysed in depth as they have not been the purpose of the study but detected once the intervention has been carried out and presented without going into detail.

The third level that benefits from this practice is the social level. In this case, improvements appear in terms of social and personal functioning [9,36,37]. These types of activities offer interesting opportunities for interaction between all participants and organizers, which encourages socialization [48], which, in turn, contributes to the protection of negative symptoms by buffering the occurrence of certain situations such as stressful ones [49]. Indeed, studies have shown that supportive social relationships help to reduce symptom severity and improve treatment response [50].

All these improvements have not only been detected through studies in which measurements have been carried out through quantitative methods, but in the case of Walburg et al. [43] and their qualitative study, it can be observed how participants are aware of the improvements they can obtain in the three levels exposed above, showing a positive attitude towards the intervention programme and their involvement in it.

However, not all the studies reviewed have shown positive results. In the case of [32], they proposed a home-based physical activity programme based on digital games controlled with body movements. Despite the participants’ prior training and the availability of complementary technical assistance, no significant changes were obtained for any of the variables studied (i.e., somatic parameters, quality of life, global assessment of functioning and social interaction). Authors such as Rice et al. [51] emphasise that proposals should not be rigid but should favour spontaneity so that the risk of failure is not high and does not negatively influence adherence. In this case, it is a programme that seems to lack this characteristic, eliminating the interaction and social support factor, which is also of great relevance for authors such as Kouvonen et al. [52], as there is no feedback between participants even from the closest environment.

This lack of improvement also occurred in the study by McGurk et al. [42], although in this case, the variable studied was cognitive functioning. They proposed a programme combining moderate aerobic physical activity with cognitive recovery exercises. However, this was a short-term intervention on a complex variable with great variability depending on the level of impairment, which could explain why the expected results were not obtained.

With regard to the second objective of this research, to identify the characteristics of the physical activity programmes considered above, as well as the barriers to their intervention, the results obtained have highlighted two points of great importance.

Considering the characteristics of the interventions analysed, all of them have intervened on the level of physical activity of the participants, although they have not operated in the same way. Although the WHO, as seen above, recommends a minimum of 150 min of physical activity and moderate to vigorous intensity, as well as aerobic activities, most of the programmes seem to have taken these recommendations into account, with the exception of the minimum time. In the case of authors such as Kerling et al. [35] and Korman et al. [41], this requirement is met or even exceeded, but in the research carried out by Brobakken et al. [39] or Areshtanab et al. [38], the established minimum is not reached. In the case of other authors such as Deenik et al. [34] or Galán et al. [40], this analysis could not be carried out due to a lack of information. However, despite all this, positive results have been obtained in terms of the increase in the rate of physical activity among the participants, probably due to the high level of sedentary lifestyle they presented and the achievement of greater adherence to physical activity thanks to the removal of certain barriers.

With regard to this second point, the barriers to be removed or reduced, it should be noted, firstly, that none of the articles analysed make direct reference to them; their identification has only been possible on the basis of the characteristics of the interventions and the variables under study. Nevertheless, a number of conclusions could be drawn. The barriers most taken into account were those related to personal (except for the side effects of medication) [20,30,33,36,37,42] and programme factors (organizational barriers and environmental constraints) [9,42,43]. This point is important as it is one of the easiest to control and directly intervenes on personal factors such as the appearance of symptoms or negative emotions that may result from these barriers, as well as on the level of motivation. This, considering the Self-Determination Theory and the Organismic Integration sub-theory [53,54] according to which there is a continuum of intrinsic–extrinsic motivation, contributes to the adherence of this type of programmes as long as the positive or self-determined types of regulation (i.e., identified, integrated and intrinsic) are promoted as opposed to the negative or non-self-determined (i.e., introjected, external and demotivated).

In the case of social factors, their presence has been almost inexistent, and they have focused exclusively on improving social support [37,40,43]. While it is true that this factor is of great relevance in contributing to the reduction of stigma towards this population sector, showing itself to be one of the barriers that most influences their recovery [55] and that allows for the improvement of self-esteem [56], it should not be forgotten that family support and the economic level of the participants also play a very important role, although they have not been taken into account in the research analysed throughout this study. The promotion of physical activity intervention programmes in standardised settings and in an inclusive manner is shown to be important.

Concerning the third objective of the study, to find the most important facilitators for adherence to physical activity programmes, it should be noted that its analysis has been marked by the lack of information about them in the studies. None of the studies considered in this work make direct reference to them, despite the fact that they are almost inevitably part of the physical activity programmes created with the aim of intervening in severe mental disorder.

However, based on the particularities of each proposal, it has been possible to draw a series of conclusions. Three facilitators clearly stand out in the results (outdoor activities, inclusion of the social component and the possibility of adapting the activities) [30,36,42,43], as they allow moments of exchange of ideas and knowledge and, therefore, the support of the participants and the possibility of improvement through the knowledge acquired. However, there are also other possibilities related to the intensity of the activities [38,39] and to more personal factors such as the development of autonomy [9,34] or social support [37,39,40]. However, these other facilitators, despite having been identified as relevant [20,23], have been difficult to identify in our analysis because they are not explicitly mentioned in the intervention programmes. This is therefore an aspect that needs to be modified, especially with a view to possible replication. It should be noted that, in the case of the study by Gyllensten and Forsberg [32], their proposal is based on the use of technological resources; however, as they are not a support but rather the means of action, they have not proved to be a facilitator.

In general, these have been attractive proposals in which a climate of trust must have predominated due to the level of adherence to the programmes, two essential facilitators already identified by authors such as Guérin et al. [20] and Chen et al. [23]. The development of all these identified facilitators contributes to an increase in the motivation of participants by achieving greater involvement in the practice of physical activity [14,19]. However, it should be noted that inclusive practices have not been detected despite also being considered a factor that promotes their practice [9].

It is important to highlight the existence of limitations in conducting this study. The first of these has been the lack of research in which the benefits on the three proposed levels (i.e., physical, psychological and social) have been studied jointly, since the research that has analysed intervention programmes is still quite limited. Moreover, no proposals have been found in which the barriers on which they intervene have been explicitly stated, making it difficult to analyse them correctly and in depth.

Taking into account all that has been exposed throughout this review, it is of interest for future research to continue with this line of study by carrying out meta-analyses that ensure greater methodological quality and obtain more solid quantitative results. It should also be noted that this research makes important practical contributions as it allows for a deeper understanding of the characteristics that physical activity-based intervention proposals should have, allowing for a better development of these and, therefore, promoting interventions that will lead to much more positive results.

## 5. Conclusions

In accordance with the results obtained, this review of the existing literature to date has made it possible to specify with greater accuracy the various benefits provided by the practice of physical activity in people with severe mental disorders, as well as the barriers and facilitators that appear in the intervention programmes proposed. In this way, it has been possible to bring together all this information in a single document with the aim of stressing the importance of planning interventions in which all these variables are taken into account.

## Figures and Tables

**Table 1 healthcare-11-01215-t001:** Analysis of studies and characteristics of interventions.

Authors (Year)	P-Participants	I-Intervention	C-Comparison	O-Results	Considered Barriers	Facilitators
Battaglia et al. [30]	Design: Longitudinal Sample: N = 18 participants Average age: 36 years Diagnosis: Schizophrenia	Anthropometric measurements, physical and mental components	**Time**:12 weeks, 2 sessions per week. Sessions of 2 h **Type**: Football-based intervention **Mode**: 20 min warm-up, 40–60 min core exercise, 10 min relaxation and stretching, 10 min feedback.	Significant reduction in weight and body mass and improvement in physical and mental components related to quality of life (*p* < 0.05).	*Personal*: Yes (physical and mental components of the quality of life) *Social*: No *Programme*: Yes (organisational level and environmental constraints)	− Outdoor development − Allows socialisation − Possibility of adaptation to personal needs
Gomes et al. [31]	Design: Longitudinal Sample: N = 19 participants (n men: 16; n women: 13) Average age: 39 years Diagnosis: Schizophrenia	Body composition, functional exercise capacity, level of physical activity, anthropometric measures, quality of life	**Time**: 2 sessions per week for 16 weeks **Type**: Indoor or outdoor sessions of moderate-vigorous intensity and aerobic group activities **Mode**: − Indoor or outdoor sessions as needed of 55–60 min duration and with an intensity of 65–75% which was increased to 75–85% in the last 8 weeks. − Aerobic type exercises based on volleyball, handball, basketball, football and walking or jogging in small teams of 2–4 participants	Reduced hip circumference and increased moderate and vigorous physical activity and environmental dominance (quality of life questionnaire variable).	*Personal*: Yes (quality of life) *Social*: No *Programme:* Yes (organisational level and environmental constraints)	− Outdoor development − Allows socialisation − Possibility of adaptation to personal needs − Moderate intensity and progressive increase
Gyllensten and Forsberg [32]	Design: Longitudinal Sample: N = 73 participants (n men: 43; n women: 30) Average age: - Diagnosis: Schizophrenia (14), neuropsychiatric disorder (14), other psychosis (4), bipolar disorders (4), other diagnoses (9), not known (28)	Somatic parameters, physical activity, quality of life, global assessment of functioning, social interaction	**Time**: 10 months **Type**: Intervention with Exergames **Mode**: Controlled games with body movements. Prior training of 4 h (2 h, 2 times) at home and availability of complementary technical assistance	No significant changes in the variables studied	*Personal*: Yes (quality of life) *Social:* No *Programme:* No	− Technological support
Firth et al. [33]	Design: Longitudinal Sample: N = 9 participants (n men: 9) Mean age: 38 years Diagnosis: SMD	Amount of physical exercise, motivation towards physical exercise, psychiatric symptoms, physical health, social functioning	**Time**: 10 weeks, 2 weekly sessions of 1 h each **Type**: Aerobic and resistance activities and varied activities of free choice **Mode**: − Session 1 of the week: 10 min warm-up, 5 min cool-down and 45 of circuit training composed of 8–16 aerobic or resistance exercise stations (push-ups, weights, sit-ups…). Each exercise lasted 1 min and a 30 s rest between each one. − Session 2 of the week: activity of the participants’ choice (group walks, gym…)	Increased physical activity was significantly correlated with reduced negative symptoms	*Personal:* Yes (motivation and symptoms) *Social:* No *Programme:* Yes (organisational level and environmental constraints)	− Outdoor realisation − Allows socialisation − Possibility of adaptation to personal needs − Moderate intensity
Mullor et al. [9]	Design: Longitudinal Sample: N = 28 participants (n men: 21; n women: 7) Mean age: 46 years Diagnosis: SMD	Anthropometric measurements, functional physical capacity, balance, coordination, social functioning and activities of daily living.	**Time**: 4 months, 3 weekly sessions of an hour and a half **Type**: Adapted gymnastics activities (aerobics, fitness and gymkhanas), cooperative games and exercises and sports activities (individual and group) **Mode**: Start with warm-up (20–25 min) and end with stretching and relaxation (10 min) and discussion-debate (5 min)	Significant improvement in those who attended more than 50% of the sessions, in terms of anthropometric measures, functional fitness, motor qualities and in social and personal functioning	*Personal:* No *Social:* No *Programme:* Yes (organisational level and environmental constraints)	− Outdoor realisation − Allows socialisation − Possibility of adapting to personal needs − Development of autonomy − Personal counselling
Deenik et al. [34]	Design: Longitudinal Sample: N = 114 participants (n men: 70; n women: 44) Mean age: 55 years Diagnosis: psychotic disorders (89), other disorders (25)	Physical activity and daily dose of medication	**Time**: 18 months, mornings and afternoons to keep participants busy **Type**: Programme based on changing habits by increasing physical activity and improving eating habits. Programme of sports activities (running, yoga, cycling…), work related activities (gardening, copy shop…), psychoeducation and training in daily life skills **Mode**: The time of waking up and the number of joint meals per day (3 meals per day) were established. Activities are carried out throughout the day, and the intensity and type of activities vary according to personal needs	Increased physical activity and reduced use of psychotropic drugs (although not directly related to increased physical activity)	*Personal:* No *Social:* No *Programme:* Yes (organisational level and environmental constraints)	− Outdoor realisation − Allows socialisation − Possibility of adapting to personal needs − Development of autonomy − Professional counselling
Kerling et al. [35]	Design: Longitudinal Sample: N = 30 participants (n men: 18; n women: 12) Mean age: 41 years Diagnosis: Major Depressive Disorder	Muscle mass	**Time**: 3 sessions per week of 45 min for 6 weeks **Type**: Moderate-intensity aerobic physical activity **Mode**: Start with 25 min on an exercise bike and end with 20 min on another machine of the user’s choice (treadmill, elliptical trainer, etc.)	Increased muscle mass	*Personal:* No *Social:* No *Programme:* Yes (organisational level and environmental constraints)	− Possibility of adapting the programme − Moderate intensity
Deenik et al. [36]	Design: Longitudinal Sample: N = 114 participants (n men: 70; n women: 44). Mean age: 55 years Diagnosis: psychotic disorders (89), other disorders (25)	Physical activity, cardiometabolic risk, psychosocial functioning, quality of life, medication and symptoms	**Time**: 18 months, mornings and afternoons to keep participants busy **Type**: Programme based on changing habits by increasing physical activity and improving eating habits. Programme of sports activities (running, yoga, cycling…), work related activities (gardening, copy shop…), psychoeducation and training in daily life skills **Mode**: The time of waking up and the number of joint meals per day (3 meals per day) were established. Activities are carried out throughout the day and the intensity (intense—moderate) and type of activities vary according to personal needs	Increased physical activity, decreased cardiometabolic risk (weight, waist circumference, systolic blood pressure and cholesterol) and improved psychosocial functioning	*Personal:* Yes (psychosocial functioning, quality of life and symptoms) *Social:* No *Programme:* Yes (level of organisational and environmental restrictions)	− Outdoor realisation − Allows socialisation − Possibility of adapting to personal needs − Development of autonomy − Professional counselling
Guerin et al. [20]	Design: Longitudinal Sample: N = 14 participants Mean age: 42 years old Diagnosis: Schizophrenia and severe bipolar disorder	Weight, body mass index, waist circumference, self-esteem, autonomy, socialisation	**Time**: 9 months **Type**: ACT (Intensive Community Treatment) approach programme **Mode**: Beginning with home exercises (e.g., walking) leading to attendance at a training centre: resistance training, aerobic, sports activities such as basketball	Reduction in: weight, body mass index and waist circumference Improvement in: self-esteem, autonomy, socialisation	*Personal:* Yes (self-esteem and autonomy) *Social:* No *Programme:* No	− Possibility of outdoor implementation − Professional advice − Socialisation − Possibility of adapting the programme − Development of autonomy
Young et al. [37]	Design: Longitudinal Sample: N = 94 participants (n men: 46; n women: 48) Mean age: 32 years Diagnosis: Schizophrenia (36), Depression (32), Anxiety (4), Bipolar disorder (8), Other disorders (8)	Depressive and anxiety symptoms, self-esteem and perceived social support	**Time**: 12 sessions with 8 jogging sessions twice a week and lasting 2 h each session; and 4 sessions of psycho-educational workshops on nutrition and body appearance lasting 2.5 h each session **Type**: Jogging sessions and psychoeducational workshops **Mode**: Jogging sessions with moderate to vigorous physical activity and psycho-educational workshops to improve body image knowledge and skills	Reduction of depressive symptoms and increase in self-esteem and social support	*Personal:* Yes (symptoms and self-esteem) *Social:* Yes (perceived social support) *Programme:* No	− Possibility of outdoor implementation − Professional advice − socialisation − Possibility of adapting the programme − Moderate intensity and progressive increase in intensity − Consideration of the support of the environment
Areshtanab et al. [38]	Design: Longitudinal Sample: N = 68 participants Mean age: 37 years Diagnosis: Schizophrenia	Quality of life, symptoms	**Time**: 24 sessions spread over 8 weeks (3 sessions per week), 12 h in total (20–40 min each session) **Type**: Outdoor aerobic exercises based on the Karvonen programme **Mode**: 5 min warm-up and 12 min of aerobic exercise such as running (65% intensity) and stretching during the first week. Increase of 2 min time and 5% intensity of core activity every 2 weeks	Reduction of symptoms and improvement of quality of life	*Personal:* Yes (quality of life and symptoms) *Social:* No *Programme:* No	− Possibility of outdoor activities − Allows socialisation − Possibility of adapting the programme − Moderate intensity and progressive increase in intensity
Brobakken et al. [39]	Design: Longitudinal Sample: N = 48 participants (n men: 28; n women: 20) Mean age: 35 years Diagnosis: Schizophrenia	Aerobic capacity and adherence to intervention	**Time**: Specialised training with 35 min sessions twice a week for 1 year. 96 sessions in total. Municipal health services with 2 h sessions **Type**: Joint intervention of specialised and municipal health service based on AIT (Aerobic Interval Training) and exercises in the clinic **Mode**: − Specialised service: 5 min warm-up and cool-down. Treadmill: 4 periods of 4 min each. Incline and speed adapted to 85–95% of each participant’s peak HR. Active rest of 3 min between each interval. 1% increase in treadmill incline to maintain intensity − Municipal health services: − Exercises in the clinic in a group setting. Positive reinforcement to maintain interest	Increased aerobic capacity and greater adherence to the intervention than the control group	*Personal:* Yes (adherence) *Social:* No *Programme:* Yes (organisational level and environmental constraints)	− Professional counselling − Allows for socialisation − Possibility of adapting the programme − Moderate intensity and progressive increase in intensity − Consideration of environmental support
Galán et al. [40]	Design: Longitudinal Sample: N = 26 participants Mean age: - Diagnosis: Severe Mental Disorder	Consideration of the programme, resources and consequences of programme participation	**Time**: From October to May on Thursdays and Fridays **Type**: Ligasame football programme. The professionals involved belong to different branches: psychologists, social workers, occupational therapists and previously trained volunteers **Modo**: 15 teams compete in a league with an active and participative methodology, as the participants are involved in the development of the league	Improvement of physical fitness, social relations, self-esteem and reduction of stress and anxiety	*Personal:* Yes (self-esteem and symptoms) *Social:* Yes (social relationships) *Programme:* Yes (organisational level and environmental constraints)	− Possibility of outdoor implementation − Professional advice − socialisation − Possibility to adapt the programme − Consideration of the support of the environment
Korman et al. [41]	Design: Longitudinal Sample: N = 42 participants (n men: 33; n women: 9) Mean age: 30 years Diagnosis: Schizophrenia	Functional exercise capacity, exercise volume, metabolic markers, psychiatric symptoms, quality of life and attitudes towards exercise	**Time**: 10 weeks, 3 times a week. Sessions of 1 h. Dietary intervention a total of 6 times (every 15 days) **Type**: Aerobic, strength and endurance training combined with a dietary intervention **Mode**: 5 min warm-up, 45 min main part and 10 min cool down	Significant improvement in functional exercise capacity, exercise volume, general psychiatric symptoms and negative psychotic symptoms No changes in anthropometric and metabolic blood markers	*Personal:* Yes (symptoms) *Social:* No *Programme:* No	− Possibility of outdoor implementation − Professional advice − socialisation − Possibility to adapt the programme
McGurk et al. [42]	Design: Longitudinal Sample: N = 34 participants (n men: 20; n women: 14) Mean age: 43 years Diagnosis: Schizophrenia and Bipolar Disorder	Physical activity and cognitive functioning	**Time**: 1 h sessions, 3 times a week for 10 weeks of cognitive recovery and 50 min sessions 3 times a week for 10 weeks of the exercise programme **Type**: Intensive cognitive recovery programme and moderate aerobic physical activity programme **Mode**: − Intensive cognitive recovery programme: cognitive exercises extracted from the Cogpack software package (version 8.0, Marker Software) − Exercise programme: 5 min warm-up, 40 min moderate aerobic exercise and 5 min cool down. Choice of activities: exercise bike, walking or running	No improvement in cognitive functioning when physical activity is implemented	*Personal:* Yes (cognitive functioning) *Social:* No *Programme:* Yes (organisational level and environmental constraints)	− Possibility of outdoor activities − Allows socialisation − Possibility of adapting the programme
Walburg et al. [43]	Design: Longitudinal Sample: N = 28 participants (n men: 9; n women: 19) Mean age: - Diagnosis: Schizophrenia, personality disorder, depression, bipolar disorder, obsessive-compulsive disorder	Assessment of the intervention	**Time**: 12 months programme with 30 sessions in total. Initial phase (6 months) with group sessions once a week. Maintenance phase (6 months) with group sessions once a month **Type**: SMILE programme: action on diet and physical activity based on the STRIDE intervention **Mode**: Sessions begin with a check-up on problems and successes from the previous session. Then, one or two scheduled topics are discussed, and the participants indicate their goals for the next session. During or at the end of the session, a 20–30 min training session (walking or indoor exercises) takes place	Positive evaluation of the intervention: positive experiences, enjoyed sharing session with people with the same goals and found the intervention useful, motivating and attractive for weight loss and achieving their goals	*Personal:* Yes (motivation, emotions and expectations) *Social:* Yes (group support) *Programme:* Yes (organisational level and environmental constraints)	− Possibility of outdoor implementation − Professional advice − Socialisation − Possibility to adapt the programme

**Table 2 healthcare-11-01215-t002:** Summary of the results.

Benefits			Intervention Characteristics	Considered Barriers			Facilitators
Physical Level	Psychological Level	Social Level		Personal	Programme	Social	
Improvement: anthropometric measures, aerobic capacity, cardiometabolic risk.	Improvement: quality of life, self-esteem, autonomy, negative symptoms of the disorder or medication.	Improvement: social functioning, personal functioning.	A minimum of 150 min Moderate to vigorous intensity Aerobic activities	Motivation, emotions, expectations, symptoms, self-esteem, autonomy	Organization Environmental constraints	Perceived social support Social relationships Group support	Outdoor activities Possibility of socialising Possibility of adaptation Professional counselling Social support Attractive proposals with a climate of trust

## Data Availability

Not applicable.

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
