# Peer review of "Benefits, Facilitators and Barrier Reductions in Physical Activity Programmes for People with Severe Mental Disorder: A Systematic Review"

_healthcare, 2023, doi:10.3390/healthcare11091215_

Round 1

Reviewer 1 Report

Good morning.

I would like to congratulate the authors for the review presented. It is a quality work, with very interesting results, which perfectly synthesise the findings in the field of study analysed. Therefore, it should be published.

In order to improve its final presentation, I propose a series of modifications.

Abstract.

The main results and conclusions should be introduced.

In abstract, one main objective and several secondary objectives are stated. In the text, three main objectives are mentioned.

Method

It is recommended to structure the Method in the following sections:

1. Search strategy: guidelines established by the PRISMA, Search terms, Databases

2. Study eligibility criteria: inclusion and exclusion criteria

3. Publication selection: criteria in the article selection process, number of articles obtained, and final number after exclusions, data extracted from the articles for the review.

Justify why the review is limited to the years 2012-2022.

Results

Start directly with the analysis of the 16 publications. Reduce the tables as much as possible so that they do not take up so many pages. Describe the main results, not present them in the discussion.

In table 1, remove the year of publication and indicate the corresponding number in the reference list (e.g., Battaglia et al.30).

Discussion

The results found should be discussed in relation to previous literature on the topic. Send the results to the previous section.

In addition to the limitations of the study, it is recommended to expand on Future research and add Practical applications.

References

In the references section, journals appearing without abbreviations should be abbreviated.

A general revision of the translation of the text is recommended.

Reviewer 2 Report

As a result of the review of this paper, the following revision is required.

1.     In the abstract, the journal search method (online or offline), the number of final selected papers, benefits, facilitators, and barriers reduction by physical activity among individulas with SMD should be described.

2.     In the research methods, the eligibility criteria must be re-described by PICO.

3.     In the study results, the characteristics of participants, intervention characteristics, and outcoms should be described in sentences as well as tables by PICO.

4.     The discussion should be based on the outcoms of the final selected previous studies and their significance. It is also necessary to discuss the quality evaluation of previous studies.

Reviewer 3 Report

It would be interesting to identify, in the characteristics of the programmes, whether they are specific to MDS, or whether people are incorporated into programmes for the whole population.
Discuss, where appropriate, the interest of developing physical activities in specific and/or normalised environments.

Reviewer 4 Report

Minor Issues

- In abstract, [lease talk more about the results found in the review. As well, please mention some important barriers and facilitators.

- In keywords, I would suggest to place physical activity before benefits, barriers …..

- In Results, I think a table with a summary of the results is needed. What are the benefits, barriers, and facilitators. Please mention all of them in a table.

- Was it possible to make a meta-analysis? If not, please add it as a suggestion for future studies. This topic is really interesting and some meta-analytic reviews are needed in the future.

Round 2

Reviewer 2 Report

This manuscript has been appropriately revised according to the reviewer's comments. Thank you for your effort